# Model Averaging for Manifold Learning

## Abstract

Manifold learning aims to extract information of high-dimensional data and provide low-dimensional representations while preserving nonlinear structures of the input data. Numerous manifold learning algorithms have been proposed in the literature. We develop a model averaging procedure to combine different manifold learning algorithms for enhancing the robustness of the result. Toward this goal, we propose a new quality metric that is tuning-free and scale-invariant by utilizing the Mahalanobis distance. The quality metric can also be used for selection of tuning parameters. Through synthetic and real data examples, we show that the new metric outperforms existing ones and the model averaging outcome provides an unanimous outcome that is always competitive with respect to visualization or classification under different contexts.

## 1 Introduction

In the era of big data, dimension reduction is a fundamental tool, which transforms high-dimensional data to low-dimensional representations while preserving the key structures of the input data. Traditional linear dimension reduction techniques, such as principal component analysis and multidimensional scaling, may be ineffective in dealing with modern complex data. Manifold learning, also known as nonlinear dimension reduction, aims to retain nonlinear structures and extract useful information of high-dimensional data in a more efficient way. Manifold learning has been an exciting research area since the seminal works by Tenenbaum et al. (2000) and Roweis & Saul (2000), which is proven to be efficient in different areas including genetic data (Moon et al., 2018; Nguyen et al., 2019), language models (Hasan & Curry, 2017; Zhao et al., 2021) and imaging analysis (Pless & Souvenir, 2009; Zhu et al., 2018), etc.

In the supervised learning context, a canonical metric often exists so that practitioners can evaluate the results from different methods. For example, the classification accuracy is the standard metric used to assess performance of a classification method. However, dimension reduction techniques are often applied as an initial step of data analysis where there is no such canonical metric, also known as the unsupervised learning. In other words, we do not have a golden rule to compare the outcomes from various manifold learning algorithms under the unsupervised context. It is then desirable to develop a strategy that can combine various manifold learning methods, the so-called model averaging for manifold learning. In contrast to model selection, model averaging is an ensemble method which has been studied extensively in the literature of regression analysis (Leamer, 1978; Hjort & Claeskens, 2003; Hansen, 2007). Practitioners may prefer averaging a number of different methods (i.e., model averaging) rather than choosing a particular one (i.e., model selection), especially when it is unclear which method is the best, e.g., under the unsupervised learning context. Different methods may have distinctive advantages in certain situations and thus their performances may vary according to specific situations. As a result, it is desirable to develop a procedure that can yield a unifying and robust outcome that is always reasonable under different contexts.

To develop model averaging for manifold learning, we need to utilize a proper quality metric. Many quality metrics for evaluating manifold learning outcomes have been proposed in the literature. Most of them are either distance-based or rank-based (Lee & Verleysen, 2009; Mokbel et al., 2013; Lee et al., 2015; Rieck & Leitte, 2017), and the others are based on particular notions (Meng et al., 2011; Zhang et al., 2012; Belov & Marik, 2021); see Section 2 for more details. These metrics may be reasonable in specific analytic contexts (Ghosh et al., 2021), while certain drawbacks hinder them from broad use. For example, the quality

metrics themselves typically involve one or multiple tuning parameters to be selected. With different tuning parameters, these metrics may lead to contradictory conclusions, which may cause confusion to practitioners.

Another notable problem of existing quality metrics of manifold learning is that they are typically based on comparing Euclidean distances and thus are not scale-invariant. These quality metrics lead to different quality assessments if the manifold learning outcomes are affine transformed. However, as the global shape features and clusters of a point cloud remains the same under affine transformation, a desirable quality metric should produce the same value under affine transformation. Moreover, a large number of manifold learning algorithms (e.g., Roweis & Saul, 2000; Belkin & Niyogi, 2003; Donoho & Grimes, 2003; Zhang & Zha, 2004) are designed to yield normalized outcomes for which the Euclidean distance is distorted. Since we intend to combine manifold learning outcomes from various scales, it is necessary to standardize the outcomes before averaging. As a result, the metrics based on the Euclidean distance is not suitable for model averaging.

The contributions of our work can be summarized as follows:

1. We develop a model averaging procedure to combine different manifold learning algorithms. Based on synthetic and real data examples, we show that the model averaging outcome always performs outstandingly with respect to visualization or classification for all considered datasets.

2. To implement the model averaging procedure, we propose a new quality metric, based on the co-ranking framework developed by Lee & Verleysen (2009), that is tuning-free and scale-invariant by utilizing the Mahalanobis distance. The proposed metric outperforms existing ones in matching the visual results and is highly positively correlated with the classification accuracy.

3. We provide a general rule of thumb for parameter tuning and selection of a manifold learning algorithm through optimizing the proposed quality metric.

To the best of our knowledge, model averaging has not been investigated in manifold learning contexts, although it has been widely studied in regression models (Raftery et al., 1997; Yin & Yuan, 2009). Zhang et al. (2013) proposed an algorithm to combine two manifold learning outcomes, while our proposed model averaging procedure provides another tool to obtain a unified outcome from various manifold learning algorithms.

The rest of the article is organized as follows. We first review model averaging and existing quality metrics in Section 2. In Section 3, we develop a model averaging procedure provided that a proper quality metric is given. In Section 4, we introduce the proposed quality metric that is particularly suitable for manifold learning model averaging. Also, we show that it can be used for tuning parameter selection. We evaluate the practical performance of the model averaging manifold learning and the validity of the proposed quality metric using both synthetic and real data in Section 5 and conclude with remarks in Section 6.

## 2 Related Work

Model averaging provides an ensemble strategy to combine multiple possible models and to reduce model uncertainty by choosing appropriate weights. The typical context of model averaging is regression analysis, for which the amount of literature is vast so that a thorough review is impossible. Bayesian and frequentist model averaging approaches are two different branches in the literature. The Bayesian approach due to its randomness on model parameters provides a coherent and systematic perspective to quantify model uncertainty (Raftery et al., 1997; Yin & Yuan, 2009). It receives extensive investigation since the seminal work by Leamer (1978), e.g., prior choices (Brock et al., 2007; Ley & Steel, 2009), dimension reduction (Green, 1995; Baragatti, 2011); see Wasserman (2000) and Fragoso et al. (2018) for comprehensive reviews. The frequentist approach has been developed much later (Hjort & Claeskens, 2003). The main strategy is to estimate the optimal weights that average out uncertainties of possible models, where the optimality is defined differently under different scenarios, see e.g., Hansen (2007); Liang et al. (2011); Hansen & Racine (2012); Zhang et al. (2016) and Zhang & Liu (2022).

Manifold learning has gained enormous popularity in high-dimensional data analysis. There are a number of quality metrics for evaluating manifold learning outcomes in the literature. Most of them can be generally

split into distance-based and rank-based categories. The idea of distance-based metrics is to see how well relative distances among data points are preserved under manifold learning. Rieck & Leitte (2017) summarized several distance-based metrics, including the average squared distance deviation with or without penalizing small deviations, and the correlation between original and embedded distances. However, it is clear that distance-based metrics are not suitable for non-isometric manifold learning algorithms, e.g., the local linear embedding. Moreover, most distance-based metrics utilize the Euclidean distance, which is not scale-invariant and thus may cause ambiguity in practice.

Rank-based metrics focus on ranks of relative Euclidean distances rather than relative Euclidean distances themselves so that they are also reasonable for non-isometric manifold learning algorithms. Lee & Verleysen (2009) utilized the co-ranking matrix to formalize a unifying framework of rank-based metrics. Specifically, they discussed the trustworthiness and continuity measures (Venna & Kaski, 2001), the local continuity meta-criterion (Chen & Buja, 2009), and the mean relative rank errors (Lee & Verleysen, 2007). Mokbel et al. (2013) proposed to reparameterize the region of interest in the co-ranking matrix. These metrics focus on different parts of the co-ranking matrix, but they all depend on the size of a neighbourhood and the Euclidean distance, i.e., they are not tuning-free nor scale-invariant. Lee et al. (2015) utilized the area under the curve in the co-ranking framework to construct a tuning-free metric.

There are other quality metrics beyond the two main categories discussed above. For example, Meng et al. (2011) proposed a global metric by exploiting the shortest path tree; Zhang et al. (2012) proposed a normalization-independent metric; Belov & Marik (2021) proposed three heuristic metrics by using simplices and geodesics. However, all these methods are computationally intensive and they require to select one or more tuning parameters.

## 3 Model Averaging

We develop the model averaging approach over multiple manifold learning outcomes, provided that a proper quality metric $\mathcal{S}$ for evaluating different manifold learning outcomes is given. Suppose that we observe an $n \times D$ high-dimensional data matrix $X = (x_1, \ldots, x_n)^\top$, where each column vector $x_i$ denotes an observation in $\mathbb{R}^D$. Further, $m$ different types of nonlinear dimension reduction techniques are applied to $X$, which yields $m$ low-dimensional representations $Y_k = (y_{1k}, \ldots, y_{nk})^\top$, with $y_{ik}$ lying in $\mathbb{R}^d$, for $k = 1, \ldots, m$. It is expected $d \ll D$ to quantify the term "dimension reduction". However, the intrinsic dimension $d$ is often unknown in practice, which can be estimated from $X$ alone (Levina & Bickel, 2004; Facco et al., 2017) or viewed as a tuning parameter. In the case of low dimensions with $d = 2$ or $3$, we can visualize $Y_k$ for better illustration.

Different manifold learning techniques typically lead to the outcomes $Y_k$ on different scales. For example, the Isomap (Tenenbaum et al., 2000) is designed to be an isometric mapping (i.e., the geodesic distances among $X$ are preserved), while the local linear embedding (LLE, Roweis & Saul, 2000) focuses on preserving local relative positions and yields normalized outcomes. To compare and combine the outcomes obtained from distinctive methods, we utilize the standardized representation,

$$z_{ik} = V_k^{-1/2}(y_{ik} - \bar{y}_k), \quad k = 1, \ldots, m \tag{1}$$

where $\bar{y}_k = \sum_{i=1}^n y_{ik}/n$ is the empirical mean vector and $V_k = \sum_{i=1}^n (y_{ik} - \bar{y}_k)(y_{ik} - \bar{y}_k)^\top/n$ is the empirical covariance matrix.

Let $Z_k = (z_{1k}, \ldots, z_{nk})^\top$. Because affine transformations preserve the global shape features and clusters of a point cloud, $Z_k$ inherits the main pattern structure of $Y_k$. It is then feasible to use $Z_k$ as a surrogate of $Y_k$. As a result, the model averaging manifold learning outcome is defined as a weighted sum of the $Z_k$'s,

$$Z_{\mathrm{ave}}(w) = \sum_{k=1}^m w_k Z_k, \tag{2}$$

where $w = (w_1, \ldots, w_m)^\top \in [0,1]^m$ satisfying $\sum_{k=1}^m w_k = 1$. To choose the optimal weights so that better outcomes receive higher weights and vice versa, it is natural to optimize the metric $\mathcal{S} \equiv \mathcal{S}\{X, Z_{\mathrm{ave}}(w)\}$ that

evaluates the quality of manifold learning outcomes,

$$w_{\mathrm{opt}} = \arg\max_{w \in [0,1]^m} \mathcal{S}\{X, Z_{\mathrm{ave}}(w)\} \quad \text{s.t.} \sum_{k=1}^{m} w_k = 1\,.$$

Using the optimal weight $w_{\mathrm{opt}}$, the model averaging manifold learning outcome is defined as $Z_{\mathrm{ave}}(w_{\mathrm{opt}})$, which overweighs better models and downweighs poorer models. In the numerical experiments, we use the built-in function fmincon with the SQP (sequential quadratic programming) algorithm in Matlab to implement the optimization above. Furthermore, we use multiple starting points to stabilize the result.

## 4  Quality Metric

It is a basic fact that a manifold is locally Euclidean, i.e., there exists a one-to-one mapping between an open subset of a Euclidean space and an open subset on the manifold. The key idea underlying almost all manifold learning algorithms is to preserve local structures of the input data $X$. Suppose that a manifold learning algorithm applied on the $n \times D$ high-dimensional $X$ produces an $n \times d$ low-dimensional representation $Y = (y_1, \ldots, y_n)^\top$. To assess the local agreement between $X$ and the manifold learning outcome $Y$, Lee & Verleysen (2009) compared the ranks of relative Euclidean distances in the input space $\mathbb{R}^D$ and the output space $\mathbb{R}^d$. However, the Euclidean distance is not scale-invariant, which means that using the Euclidean distance leads to different quality assessments if $Y$ is affine transformed. As affine transformations do not change the global shape features and clusters of a point cloud, a quality metric should produce the same value for the affine transformed $Y$. The scale-invariance of the quality metric is particularly required for model averaging, as the model averaging manifold outcome is defined by the standardized outcomes. In terms of data visualization, the affine transformed $Y$ with the correspondingly adjusted axis scales can produce exactly the same figure as the original $Y$. Therefore, it is not suitable to compare the ranks based on the Euclidean distance. Instead, we propose to use the Mahalanobis distance, which is scale-invariant.

Specifically, let $\bar{x} = \sum_{i=1}^{n} x_i / n$ and $\bar{y} = \sum_{i=1}^{n} y_i / n$, and the empirical covariance matrices of $X$ and $Y$ are given by

$$V_X = \frac{\sum_{i=1}^{n}(x_i - \bar{x})(x_i - \bar{x})^\top}{n} \text{ and } V_Y = \frac{\sum_{i=1}^{n}(y_i - \bar{y})(y_i - \bar{y})^\top}{n}\,,$$

respectively. The Mahalanobis distance between $x_i$ and $x_j$ in $\mathbb{R}^D$ is defined as

$$\delta_{ij}^X = \delta^X(x_i, x_j) = \sqrt{(x_i - x_j)^\top V_X^{-1}(x_i - x_j)}$$

and that between $y_i$ and $y_j$ in $\mathbb{R}^d$ is

$$\delta_{ij}^Y = \delta^Y(y_i, y_j) = \sqrt{(y_i - y_j)^\top V_Y^{-1}(y_i - y_j)}\,. \tag{3}$$

In the case that $V_X$ is singular (e.g., $D > n$), we use the diagonal variance matrix $\tilde{V}_X = \mathrm{diag}(V_X)$ to replace $V_X$. The cases of $d > n$ are rarely observed in practice, and thus we assume that $V_Y$ is always invertible.

The rank of $x_j$ with respect to $x_i$ is defined as

$$\rho_{ij}^X = |\{k : \delta_{ik}^X < \delta_{ij}^X \text{ or } (\delta_{ik}^X = \delta_{ij}^X \text{ and } 1 \le k < j \le n)\}|\,,$$

where $|\{\cdot\}|$ denotes the cardinality of the set $\{\cdot\}$. The condition $(\delta_{ik}^X = \delta_{ij}^X \text{ and } 1 \le k < j \le n)$ is to ensure that the ranks $\rho_{ij}^X$, for $j = 1, \ldots, n$, are unique. Similarly, the rank of $y_j$ with respect to $y_i$ is defined as

$$\rho_{ij}^Y = |\{k : \delta_{ik}^Y < \delta_{ij}^Y \text{ or } (\delta_{ik}^Y = \delta_{ij}^Y \text{ and } 1 \le k < j \le n)\}|\,.$$

For given $i$ and $K$, the common indices of $\nu_{i,K}^X = \{j : 1 \le \rho_{ij}^X \le K\}$ and $\nu_{i,K}^Y = \{j : 1 \le \rho_{ij}^Y \le K\}$ reflect how well the local $K$-nearest neighbours of the individual $i$ are preserved by the manifold learning algorithm.

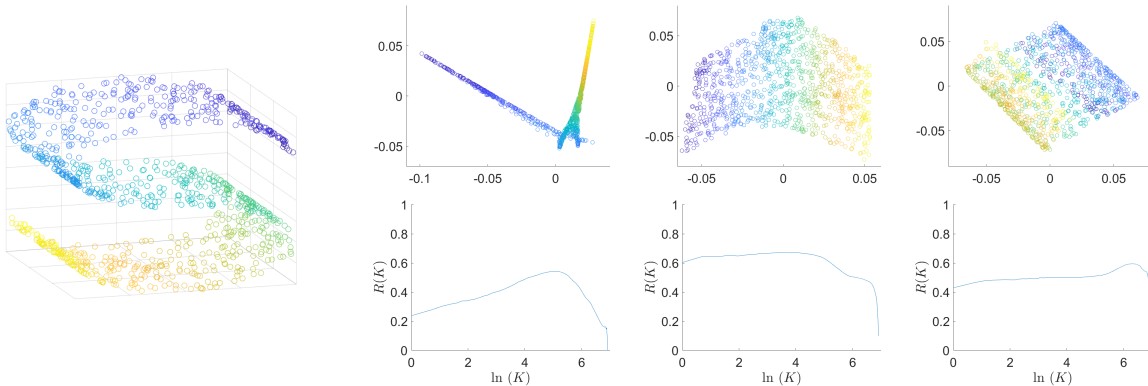

Figure 1: Left panel: the original dataset of the $S$ shape; the top row of right panel: the manifold learning outcomes using LLE with the neighbourhood size $K_{\mathrm{LLE}} = 5$ (left), 15 (middle) and 50 (right); the bottom row of right panel: the corresponding curves $(\ln(K), R(K))$ with $\mathcal{S} = 2.66$ (left), 4.24 (middle) and 3.48 (right).

Therefore, the value $|\nu^X_{i,K} \cap \nu^Y_{i,K}|$ can be used as a quality score of the manifold learning algorithm for the $i$-th observation. The averaged score for all individuals is then defined as

$$Q(K) = \frac{\sum_{i=1}^n |\nu^X_{i,K} \cap \nu^Y_{i,K}|}{Kn},$$

for $K = 1, \ldots, n-1$. However, as $Q(K)$ depends on $K$, it is not favored as our quality metric. To fairly average out the effects of different values of $K$, we note that a random coordinate of $X$ leads to $Q(K) \approx K/(n-1)$ (Chen & Buja, 2009), and thus we propose the rescaled version of $Q(K)$ as

$$R(K) = \frac{(n-1)Q(K) - K}{n - 1 - K},$$

for $K = 1, \ldots, n-2$. For different values of $K$, $R(K)$ has the same range from zero (random embedding, e.g., a random permutation) to one (perfect embedding, e.g., a rigid transformation including rotation and translation).

Finally, to produce a global quality score that avoids choosing a particular $K$, we view $R(K)$ as a continuous function of $K$ and define the tuning-free quality metric as the area under the curve (AUC) of $R(K)$ using the log scale,

$$\mathcal{S}(X, Y) = \int_1^{n-2} R(K) \, d\ln(K). \tag{4}$$

The reason of using a log-scaled axis of $K$ is that the rank preservation for a large neighbourhood is less important than that for a small neighbourhood (Lee et al., 2015), because the basis of manifold learning is the preservation of local structures. An important property of $\mathcal{S}$ is that $\mathcal{S}(X, Y) = \mathcal{S}(X, YA + b^\top)$, for any invertible matrix $A \in \mathbb{R}^{d \times d}$ and vector $b \in \mathbb{R}^d$, i.e., it is scale-invariant due to utilizing the Mahalanobis distance. In particular, we have $\mathcal{S}(X, Y) = \mathcal{S}(X, Z)$, where $Z$ is the standardized version of $Y$. Therefore, it is suitable to be used for the model averaging purpose.

Most of the manifold learning algorithms require one or more tuning parameters to be chosen. For example, in the $k$-nearest neighbours ($k$-NN) based methods (e.g., Tenenbaum et al., 2000; Roweis & Saul, 2000; McInnes et al., 2018; Budninskiy et al., 2019; Tan et al., 2023), it is a nontrivial task to choose a proper $k$ for quantifying the size of the local structure in order to approximate the unknown manifold. Different tuning parameters may lead to substantially different manifold learning outcomes, some of which could be even misleading. Since the quality metric $\mathcal{S}$ itself is tuning-free and acts as a quality score of a manifold learning outcome, we can maximize $\mathcal{S}$ with respect to the tuning parameter for any given manifold learning

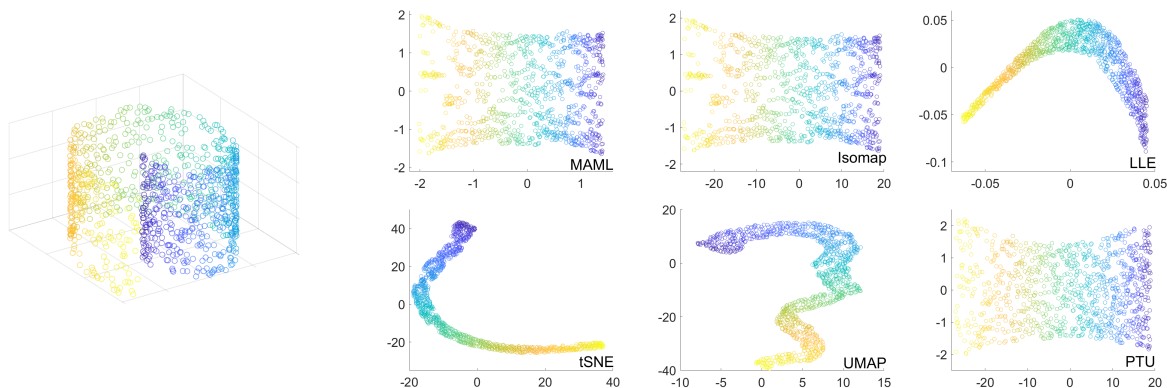

Figure 2: Left: the original dataset of the Swiss roll; right: the corresponding manifold learning outcomes from all the six algorithms.

algorithm. This provides a general rule of thumb to select the tuning parameter for a manifold learning algorithm.

As an illustration of tuning parameter selection, we apply the LLE (Roweis & Saul, 2000) to a synthetic dataset and compute the corresponding curves $(\ln(K), R(K))$, $K \in [1, n-2]$. Specifically, we generate an $S$ shape dataset (see Section 5 for details) and apply LLE with the neighbourhood size (the tuning parameter) $K_{\mathrm{LLE}} = 5$, 15 and 50. From the results shown in Figure 1, we see that the manifold learning outcome with $K_{\mathrm{LLE}} = 15$ performs the best visually, followed by the cases with $K_{\mathrm{LLE}} = 50$ and $K_{\mathrm{LLE}} = 5$. Correspondingly, the value $\mathcal{S} = 4.24$ with $K_{\mathrm{LLE}} = 15$ is indeed the largest; the other values of $\mathcal{S}$ are 3.48 and 2.66 for $K_{\mathrm{LLE}} = 50$ and $K_{\mathrm{LLE}} = 5$, respectively. We conclude from this example that a larger value of $\mathcal{S}$ corresponds to an outcome with a better visual presentation, and thus maximizing $\mathcal{S}$ with respect to the tuning parameter is a reasonable rule to select the tuning parameter.

## 5 Experiments

We examine the practical performance of the model averaging manifold learning (MAML) through numerical experiments. We adopt several well-known manifold learning algorithms as the baseline candidates, including Isomap (Tenenbaum et al., 2000), LLE (Roweis & Saul, 2000), tSNE (Van der Maaten & Hinton, 2008), UMAP (McInnes et al., 2018) and PTU (Budninskiy et al., 2019). All the aforementioned algorithms contain their own tuning parameters to be chosen. For each of these algorithms, we select the corresponding tuning parameter by maximizing $\mathcal{S}$. We then follow the model averaging procedure described in Section 3 to obtain the MAML outcome.

For clarity, we use $\mathcal{S}(\mathrm{Mahalanobis})$ to denote the proposed quality metric using the Mahalanobis distance. As a comparison with the proposed quality metric $\mathcal{S}(\mathrm{Mahalanobis})$, we further consider the following two tuning-free metrics: (i) the version of $\mathcal{S}(\mathrm{Mahalanobis})$ with the Mahalanobis distance replaced by the Euclidean distance (Lee et al., 2015), denoted by $\mathcal{S}(\mathrm{Euclidean})$; (ii) the average residual variance (Rieck & Leitte, 2017), $\mathcal{R} = 1 - \sum_{i=1}^{n} v_i/n$, where $v_i = [\mathrm{corr}(\{d_{i1}, \ldots, d_{in}\}, \{\delta_{i1}, \ldots, \delta_{in}\})]^2$ with $d_{ij} = \sqrt{(x_i - x_j)^\top (x_i - x_j)}$ and $\delta_{ij} = \sqrt{(y_i - y_j)^\top (y_i - y_j)}$. Larger values of $\mathcal{S}(\mathrm{Mahalanobis})$ and $\mathcal{S}(\mathrm{Euclidean})$ and lower values of $\mathcal{R}$ indicate better quality.

### 5.1 Synthetic Data

We generate three surface datasets in $\mathbb{R}^3$ and apply the aforementioned algorithms including the MAML to produce 2-dimensional outcomes. We fix the sample size $n = 1000$. The synthetic data $X = (x_1, \ldots, x_n)^\top$ include:

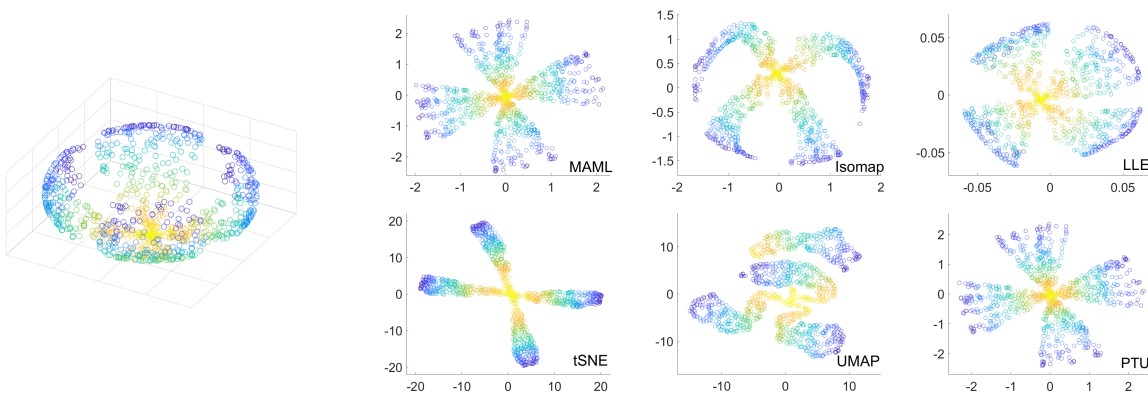

Figure 3: Left panel: the original dataset of the four-petal shape; right panel: the corresponding manifold learning outcomes from all the six algorithms in comparison.

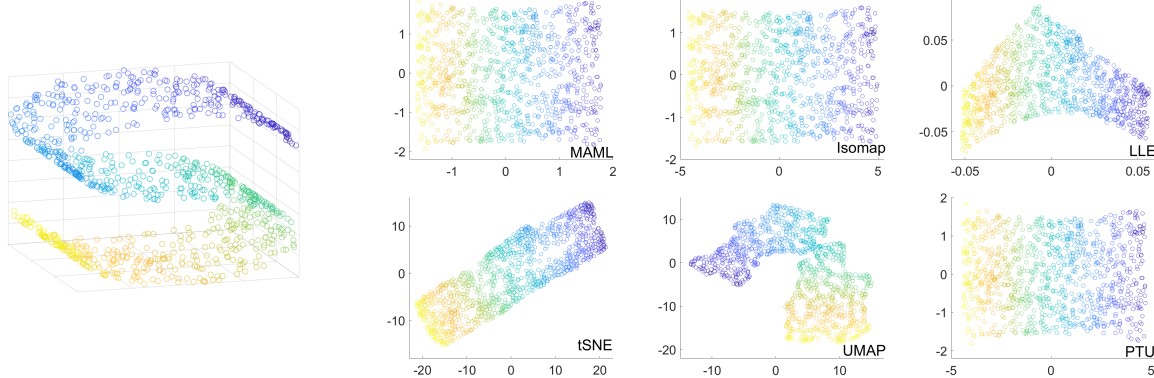

Figure 4: Left panel: the original dataset of the $S$ shape; right panel: the corresponding manifold learning outcomes from all the six algorithms in comparison.

1. Swiss roll: $x_i = \big(u_{i1}\cos(u_{i1}), u_{i1}\sin(u_{i1}), u_{i2}\big)^\top$, where $3 = u_{11} < \cdots < u_{n1} = 10$ are equidistant between 3 and 10, and $u_{i2}$ is from a uniform distribution on $[0, 3]$, i.e., $u_{i2} \overset{\text{i.i.d.}}{\sim} U[0, 3]$;

2. Four-petal shape: $x_i = \big(\sin(\theta_i)\cos(\phi_i), \sin(\theta_i)\sin(\phi_i), \cos(\theta_i)\big)^\top$, where $\theta_i \overset{\text{i.i.d.}}{\sim} U[\pi/4, \pi]$ and $\phi_i$ is drawn from one of the four distributions: $U[0, \pi/3], U[\pi/2, 5\pi/6], U[\pi, 4\pi/3]$ and $U[3\pi/2, 11\pi/6]$, each of which produces a sample of size $n/4$;

3. $S$ shape: $x_i = \big(\sin(u_{i1}), u_{i2}, \text{sign}(u_{i1})\{\cos(u_{i1}) - 1\}\big)^\top$, where $u_{i1} \overset{\text{i.i.d.}}{\sim} U[-3\pi/2, 3\pi/2]$ and $u_{i2} \overset{\text{i.i.d.}}{\sim} U[1, 4]$.

For the three synthetic datasets, the visual results are shown in Figures 2 to 4 respectively. For the Swiss roll data as shown in Figure 2, Isomap and PTU perform the best in unwrapping the surface among the five candidate algorithms, while MAML yields almost the same result as Isomap, demonstrating that the model averaging procedure can pick up the best candidate. Similar observations can be made for the $S$ shape data in Figure 4: MAML, Isomap and PTU all produce the rectangular coordinates that embed the $S$ shape into $\mathbb{R}^2$ with minimal distortion. In Figure 3, MAML produces almost the same result as PTU whose outcome accurately preserves the shape of the four petals, while Isomap, LLE and tSNE deliver four petals but the leaves are distorted. Note that the four petals of the original surface are identical and symmetric. The optimal weights $w_{\text{opt}}$ for computing MAML are given in Table 2. In summary, MAML always leads to the outcome closest to the best visual result produced by one of the candidate algorithms, which corroborates the robustness of the model averaging approach.

Table 1: The values of the three quality metrics, the proposed metric using the Mahalanobis distance $\mathcal{S}$(Mahalanobis), the corresponding version using the Euclidean distance $\mathcal{S}$(Euclidean), and the average residual variance $\mathcal{R}$, with the best value highlighted in boldface under the six algorithms for the three synthetic datasets.

| Quality metrics | Datasets | MAML | Isomap | LLE | tSNE | UMAP | PTU |
|---|---|---|---|---|---|---|---|
| $\mathcal{S}$(Mahalanobis) | Swiss roll | **4.50** | **4.50** | 3.47 | 2.86 | 2.99 | 4.34 |
| | Four-petal shape | 6.00 | 5.17 | 5.00 | 5.13 | 4.43 | **6.06** |
| | $S$ shape | 4.40 | **4.41** | 4.10 | 4.33 | 4.25 | 4.39 |
| $\mathcal{S}$(Euclidean) | Swiss roll | 2.28 | 5.63 | 3.65 | 5.36 | 5.31 | **5.96** |
| | Four-petal shape | 5.72 | 5.24 | 5.00 | 5.33 | 4.53 | **6.07** |
| | $S$ shape | 4.39 | 5.96 | 4.20 | 5.74 | 4.93 | **6.19** |
| $\mathcal{R}$ | Swiss roll | 0.78 | 0.47 | 0.52 | 0.43 | **0.41** | 0.47 |
| | Four-petal shape | 0.20 | 0.19 | 0.19 | 0.39 | 0.58 | **0.18** |
| | $S$ shape | 0.28 | 0.27 | 0.43 | 0.27 | 0.43 | **0.26** |

Table 2: The optimal weights $w_{\mathrm{opt}}$ for computing MAML from the five candidate algorithms for the three synthetic datasets.

| Datasets | Isomap | LLE | tSNE | UMAP | PTU |
|---|---|---|---|---|---|
| Swiss roll | 1 | 0 | 0 | 0 | 0 |
| Four-petal shape | 0 | 0.091 | 0 | 0.008 | 0.901 |
| $S$ shape | 0.513 | 0.002 | 0.003 | 0 | 0.483 |

In Table 1, we show the three quality metric scores for all six algorithms computed from each of the synthetic surface datasets. We see that the values of $\mathcal{S}$(Mahalanobis) of MAML are always close to the highest values of $\mathcal{S}$(Mahalanobis) among the five candidate models. Combining the values of $\mathcal{S}$(Mahalanobis) in Table 1 with the visual exhibition results in Figures 2–4, we conclude that the metric $\mathcal{S}$(Mahalanobis) indeed reflects the quality of data visualization of manifold learning outcomes: A larger value of $\mathcal{S}$(Mahalanobis) corresponds to a manifold learning outcome that unwraps the surface with less distortion.

On the other hand, the quality metric $\mathcal{R}$ often does not correctly reflect the visual quality. For example, UMAP and tSNE associated with the worst visual results nevertheless yield the best two values of $\mathcal{R}$ for the Swiss roll data. As to the quality metric $\mathcal{S}$(Euclidean), it yields quite different values for visually almost the same manifold learning outcomes (whose scales are different). For example, the results of MAML are basically rescaled versions of those of Isomap for the Swiss roll and $S$ shape data, whereas the corresponding values of $\mathcal{S}$(Euclidean) are distinct. In contrast, the corresponding values of $\mathcal{S}$(Mahalanobis) are almost the same, i.e., it correctly match visual results. Recall that $\mathcal{S}$(Mahalanobis) utilizes the Mahalanobis distance, which is scale-invariant. This demonstrates the importance of the scale-invariance property in assessing manifold learning outcomes.

## 5.2 Real data applications

We further apply MAML as well as the other five candidate algorithms to the following three real image datasets:

1. COIL-20 (Nene et al., 1996) is a dataset including 1440 grayscale images of 20 objects from different shooting angles. Each image consists of $128 \times 128$ pixels.

2. MNIST (LeCun, 1998) is a dataset including 70000 grayscale images of handwritten digits (0 to 9). We randomly select a subsample of size 2000 as an illustration. Each image consists of $28 \times 28$ pixels.

3. Chest X-ray (Kermany et al., 2018) is a dataset including 5,863 grayscale images of patient chest X-rays. The images are classified as normal or pneumonic. We rescale the images as $10^3 \times 10^3$ pixels and randomly select 1000 images (half normal and half pneumonic) as an illustration.

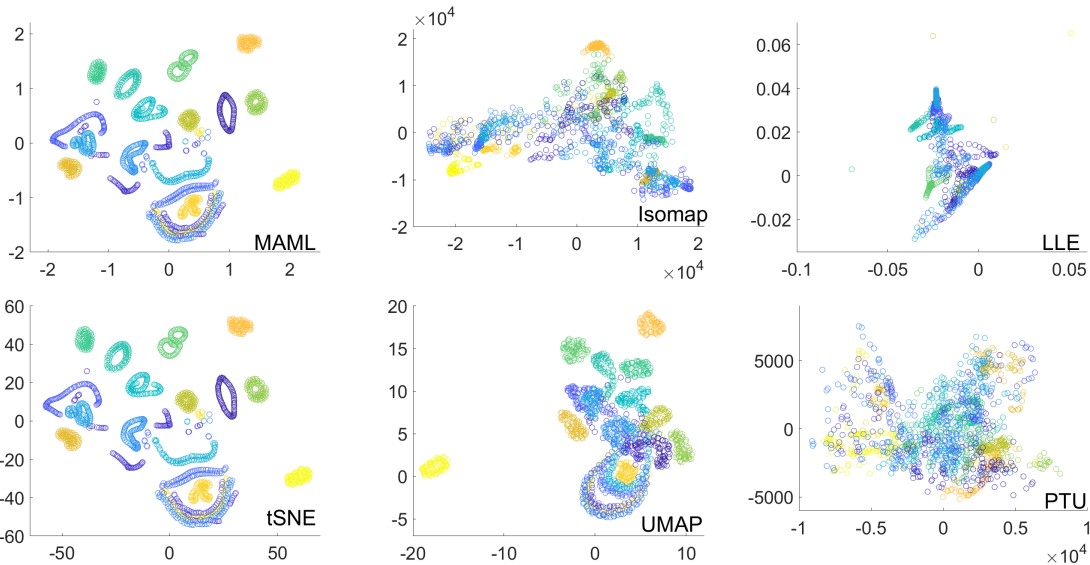

Figure 5: The manifold learning outcomes with $d = 2$ for the COIL-20 dataset. Each color corresponds to one object.

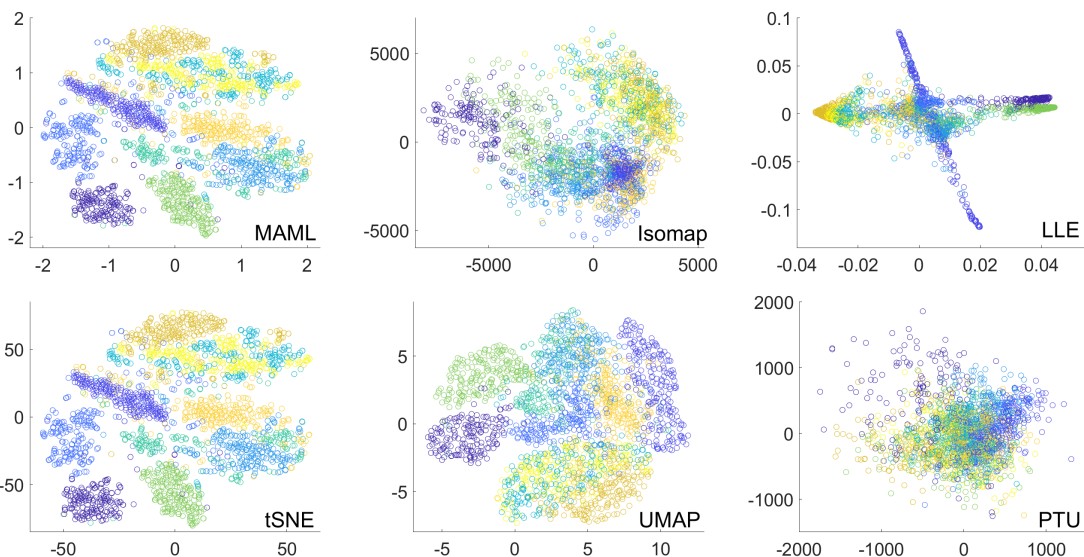

Figure 6: The manifold learning outcomes with $d = 2$ for the MNIST dataset. Each color corresponds to one digit.

For each image, the input data are treated as a vector of length equal to the number of pixels. For data visualization, we apply all the algorithms with $d = 2$ and show the outcomes in Figures 5–7. From all of the figures, we see that the outcomes of MAML are approximately rescaled versions of those of tSNE, whose outcomes can achieve the best visual separation of different classes among the candidate algorithms. Such observation is consistent with that in Section 5.1, demonstrating that MAML always picks up the candidate algorithm with the best visual result.

To evaluate the performances of different manifold learning algorithms in the supervised learning, we further consider the classification task based on the manifold learning outcomes, treating the objects in the COIL-20 dataset, the digits in the MNIST dataset and the disease status in the chest X-ray dataset as the class labels, respectively. Since the intrinsic dimension $d = 2$ may result in too much loss of information for image data,

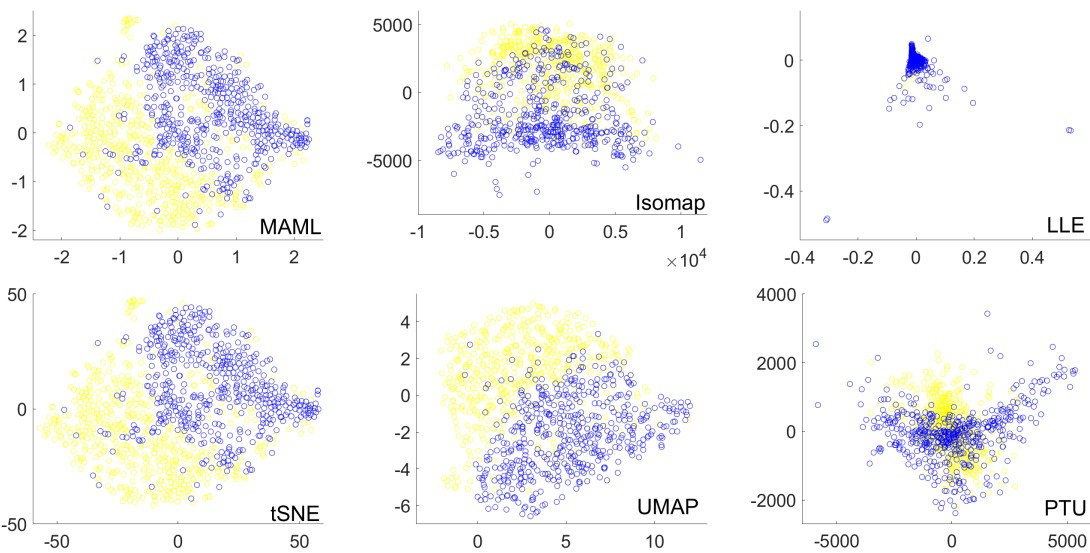

Figure 7: The manifold learning outcomes with $d = 2$ for the chest X-ray dataset. The normal and pneumonic individuals are marked in yellow and blue, respectively.

we also apply the manifold learning algorithms with $d$ estimated by the method proposed by Facco et al. (2017). The idea in Facco et al. (2017) is to exploit the relationship between the intrinsic dimension and the ratio between the first and second shortest distances for a given point. The estimate of $d$ is given by the slope of an estimated straight line. The method leads to $d = 8$ for the COIL-20 dataset, $d = 13$ for the MNIST dataset and $d = 21$ for the chest X-ray dataset. For a given dataset, we first apply all the manifold learning algorithms to the whole dataset to obtain the manifold learning outcomes. Then we randomly select 80% of the data as the training set and the rest as the test set, and we use a vanilla $k$-NN classifier for all manifold learning outcomes with different values of $k$. The procedure is repeated 100 times and the classification accuracy results are summarized in Table 3.

In terms of the mean classification accuracy rate, Table 3 shows that MAML and tSNE perform similarly and both outperform others for the COIL-20 dataset under the cases with $d = 2$ and 8, which is also true for the MNIST dataset with $d = 2$. This coincides with our visual observation in Figures 5 and 6. In the case with $d = 13$ for the MNIST dataset, the differences between the results using different methods are small and all methods improve significantly compared with those with $d = 2$; in particular, UMAP performs the best yet with only minor advantages over others. For the chest X-ray dataset, UMAP performs the best overall, while the advantage is minimal under $d = 21$; MAML and tSNE perform similarly; the performances of Isomap, LLE and PTU improve greatly from $d = 2$ to $d = 21$. Table 4 shows the optimal weights $w_{\text{opt}}$ for computing MAML under all cases. MAML always assigns the highest weight to tSNE except for the MNIST dataset with $d = 2$, for which it assigns the highest weight to Isomap. As seen in Table 3, tSNE indeed performs outstandingly among the five candidate algorithms, especially when $d = 2$. Overall, we conclude that MAML is robust and always performs competitively compared with the best candidate algorithm, which, however, may differ for different datasets.

In Table 5, we show the values of the three quality metrics of all algorithms for the three image datasets. Comparing Table 5 with Table 3, we see that classification performances of MAML and the candidate algorithm that produces the highest $\mathcal{S}(\text{Mahalanobis})$ are very similar. Moreover, the value of $\mathcal{S}(\text{Mahalanobis})$ and the classification accuracy are highly positively correlated. For example, for the MNIST dataset with $d = 2$, the rank of values of $\mathcal{S}(\text{Mahalanobis})$ is exactly the same as that of the classification results among the five candidate algorithms. Similar observation can be seen for $\mathcal{S}(\text{Euclidean})$. However, the quality metric $\mathcal{R}$ favours Isomap or PTU for all cases, but these two algorithms often perform worse than others in terms of classification. Specifically, for a given quality metric and a dataset with $d = 2$, we compute the correlation between the quality metric values and the classification accuracy averaged over $k$. The correlation

Table 3: The mean (standard deviation) of the classification accuracy rate based on the manifold learning outcomes. The best results are highlighted in boldface.

| Datasets | $d$ | $k$ | MAML | Isomap | LLE | tSNE | UMAP | PTU |
|---|---|---|---|---|---|---|---|---|
| COIL-20 | 2 | 10 | **0.93** (0.01) | 0.68 (0.02) | 0.75 (0.02) | **0.93** (0.02) | 0.83 (0.02) | 0.49 (0.03) |
| | | 20 | **0.89** (0.02) | 0.66 (0.03) | 0.73 (0.02) | **0.89** (0.02) | 0.81 (0.02) | 0.48 (0.03) |
| | | 40 | **0.87** (0.02) | 0.61 (0.03) | 0.69 (0.02) | 0.86 (0.02) | 0.75 (0.02) | 0.43 (0.03) |
| | | 80 | 0.78 (0.03) | 0.54 (0.03) | 0.64 (0.03) | **0.79** (0.03) | 0.71 (0.02) | 0.39 (0.03) |
| | 8 | 10 | 0.95 (0.01) | 0.86 (0.02) | 0.84 (0.02) | **0.96** (0.02) | 0.92 (0.02) | 0.87 (0.02) |
| | | 20 | **0.93** (0.02) | 0.82 (0.02) | 0.80 (0.02) | 0.92 (0.02) | 0.86 (0.02) | 0.80 (0.02) |
| | | 40 | **0.90** (0.02) | 0.75 (0.03) | 0.76 (0.02) | **0.90** (0.02) | 0.82 (0.02) | 0.73 (0.03) |
| | | 80 | **0.85** (0.02) | 0.64 (0.03) | 0.72 (0.02) | 0.84 (0.02) | 0.77 (0.03) | 0.61 (0.03) |
| MNIST | 2 | 10 | **0.86** (0.02) | 0.50 (0.02) | 0.65 (0.02) | **0.86** (0.02) | 0.73 (0.02) | 0.28 (0.02) |
| | | 20 | 0.82 (0.02) | 0.51 (0.02) | 0.65 (0.02) | **0.83** (0.02) | 0.73 (0.02) | 0.29 (0.02) |
| | | 40 | 0.78 (0.02) | 0.53 (0.02) | 0.64 (0.02) | **0.79** (0.02) | 0.73 (0.02) | 0.30 (0.02) |
| | | 80 | 0.73 (0.02) | 0.52 (0.02) | 0.61 (0.02) | **0.75** (0.02) | 0.72 (0.02) | 0.30 (0.02) |
| | 13 | 10 | 0.85 (0.02) | 0.86 (0.02) | 0.86 (0.02) | 0.88 (0.02) | **0.89** (0.01) | 0.80 (0.02) |
| | | 20 | 0.83 (0.02) | 0.84 (0.02) | 0.85 (0.02) | 0.87 (0.02) | **0.88** (0.01) | 0.77 (0.02) |
| | | 40 | 0.81 (0.02) | 0.82 (0.02) | 0.83 (0.02) | 0.85 (0.02) | **0.87** (0.01) | 0.74 (0.02) |
| | | 80 | 0.78 (0.02) | 0.79 (0.02) | 0.80 (0.02) | 0.84 (0.02) | **0.85** (0.02) | 0.70 (0.02) |
| Chest X-ray | 2 | 10 | **0.89** (0.02) | 0.77 (0.02) | 0.73 (0.03) | **0.89** (0.02) | **0.89** (0.02) | 0.70 (0.03) |
| | | 20 | **0.88** (0.02) | 0.77 (0.02) | 0.72 (0.03) | **0.88** (0.02) | **0.88** (0.02) | 0.71 (0.03) |
| | | 40 | 0.86 (0.02) | 0.78 (0.02) | 0.73 (0.03) | 0.86 (0.02) | **0.88** (0.02) | 0.70 (0.03) |
| | | 80 | 0.85 (0.02) | 0.78 (0.02) | 0.72 (0.03) | 0.86 (0.02) | **0.87** (0.02) | 0.70 (0.03) |
| | 21 | 10 | 0.88 (0.02) | **0.91** (0.02) | **0.91** (0.02) | 0.88 (0.02) | **0.91** (0.02) | **0.91** (0.02) |
| | | 20 | 0.88 (0.02) | **0.91** (0.01) | 0.90 (0.02) | 0.87 (0.02) | **0.91** (0.02) | **0.91** (0.02) |
| | | 40 | 0.88 (0.02) | **0.90** (0.02) | 0.89 (0.02) | 0.86 (0.02) | **0.90** (0.02) | **0.90** (0.02) |
| | | 80 | 0.88 (0.02) | 0.88 (0.02) | 0.84 (0.03) | 0.85 (0.02) | **0.89** (0.02) | **0.89** (0.02) |

Table 4: The optimal weights $w_{\text{opt}}$ for computing MAML from the five candidate algorithms for the three real image datasets.

| Datasets | $d$ | Isomap | LLE | tSNE | UMAP | PTU |
|---|---|---|---|---|---|---|
| COIL-20 | 2 | 0 | 0.014 | 0.986 | 0 | 0 |
| | 8 | 0 | 0.043 | 0.943 | 0 | 0.014 |
| MNIST | 2 | 0.004 | 0.001 | 0.978 | 0.011 | 0.006 |
| | 13 | 0.965 | 0 | 0 | 0.035 | 0 |
| Chest X-ray | 2 | 0 | 0 | 1 | 0 | 0 |
| | 21 | 0 | 0.006 | 0.979 | 0.015 | 0 |

of $\mathcal{S}$(Mahalanobis) for the datasets COIL-20, MINST, and Chest X-ray are 0.94, 0.89, and 0.79, respectively; the corresponding values of $\mathcal{S}$(Euclidean) are 0.92, 0.89, and 0.81, respectively; while corresponding values of $\mathcal{R}$ are 0.21, -0.21, and -0.1, respectively. Finally, we note that a higher value of $\mathcal{S}$(Mahalanobis) does not *always* leads to a better classification result, which is as expected – since any quality metric only assesses the structural preservation of the manifold learning outcome and does not use any label information.

## 6 Discussion

We develop a model averaging procedure to combine different manifold learning algorithms and produce a unified and robust outcome. Toward this goal, we propose a new quality metric that is tuning-free and scale-invariant. The model averaging procedure shows the desired numerical performance in that it mimics the candidate algorithm which yields the best visualization or classification accuracy. Comparing to the existing quality metrics, our proposed metric matches visual results due to its scale-invariance property and is often

Table 5: The values of the three quality metrics, the proposed metric using the Mahalanobis distance $\mathcal{S}$(Mahalanobis), the version using the Euclidean distance $\mathcal{S}$(Euclidean), and the average residual variance $\mathcal{R}$, under the six algorithms for the three real image datasets. The best results are highlighted in boldface.

| Quality metrics | Datasets | $d$ | MAML | Isomap | LLE | tSNE | UMAP | PTU |
|---|---|---|---|---|---|---|---|---|
| $\mathcal{S}$(Mahalanobis) | COIL-20 | 2 | **4.37** | 2.68 | 2.61 | **4.37** | 3.22 | 1.89 |
| | | 8 | **3.88** | 3.86 | 2.82 | 3.87 | 3.49 | 3.57 |
| | MNIST | 2 | **2.55** | 1.07 | 1.14 | 2.54 | 1.61 | 0.64 |
| | | 13 | **2.97** | **2.97** | 1.61 | 2.62 | 2.43 | 2.58 |
| | Chest X-ray | 2 | **2.46** | 1.47 | 1.31 | **2.46** | 1.64 | 1.34 |
| | | 21 | **2.87** | 2.51 | 2.47 | **2.87** | 2.42 | 2.60 |
| $\mathcal{S}$(Euclidean) | COIL-20 | 2 | 4.65 | 3.00 | 2.69 | **4.73** | 3.45 | 2.09 |
| | | 8 | 4.09 | 4.65 | 3.05 | **4.92** | 4.31 | 4.49 |
| | MNIST | 2 | 3.30 | 1.42 | 1.49 | **3.40** | 2.09 | 0.85 |
| | | 13 | 3.87 | **4.38** | 2.14 | 3.69 | 3.75 | 3.81 |
| | Chest X-ray | 2 | 2.76 | 1.60 | 1.37 | **2.77** | 1.85 | 1.47 |
| | | 21 | 3.14 | 3.76 | 2.74 | 3.58 | 3.53 | **4.28** |
| $\mathcal{R}$ | COIL-20 | 2 | 0.57 | **0.41** | 0.82 | 0.56 | 0.70 | 0.55 |
| | | 8 | 0.72 | 0.24 | 0.83 | 0.59 | 0.68 | **0.23** |
| | MNIST | 2 | 0.74 | **0.70** | 0.92 | 0.73 | 0.80 | 0.82 |
| | | 13 | 0.38 | **0.25** | 0.93 | 0.66 | 0.59 | 0.32 |
| | Chest X-ray | 2 | 0.64 | 0.50 | 0.94 | 0.65 | 0.59 | **0.48** |
| | | 21 | 0.81 | 0.21 | 0.65 | 0.42 | 0.40 | **0.11** |

positively correlated to the classification accuracy, indicating its validity. The proposed quality metric can also be used for selection of tuning parameters.

The theoretical properties of the proposed quality metric, as well as those in the literature, have not been investigated. An important but difficult question is to which type of manifold a quality metric is suitable. As the intrinsic dimension is often unknown and chosen manually, it warrants further research on possible combination of manifold learning outcomes of different dimensions.

## Acknowledgement

We are grateful to four anonymous reviewers for their helpful and constructive comments that have greatly improved the paper.

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
