# OpenReview forum: "Model Averaging for Manifold Learning"
_TMLR — Rejected by TMLR_

### Review · Reviewer_R2Fj · 2023-04-04

**Summary Of Contributions:**

In the paper, the authors proposed a new metric to measure the quality of the manifold learning method. The method outperforms other methods in that it is tuning-free and scale-invariant.  Using the proposed metric, the authors proposed a model averaging procedure to
combine different manifold learning algorithms. Results show that the model averaging outcome always performs similar to the candidate algorithm that yields the best visualization or classification accuracy.

**Audience:**

Yes

**Claims And Evidence:**

No

**Requested Changes:**

1. Conduct experiments to compare the proposed metric with other related metrics.
2. The statement "A larger value of S usually corresponds to a manifold learning outcome with a better visualization" is not convincing. It is better to conduct an experiment that involves some people to vote on the best visualization. Alternatively, the authors can select other datasets that show a clearer difference.


**Strengths And Weaknesses:**

Strengths:
1. The new metric is tuning-free and scale-invariant.

Weaknesses:
1. One of the paper's main contributions is a new tuning-free and scale-invariant metric for manifold learning. However, the paper lacks a comparison between the proposed metric and other related metrics. This makes it difficult to assess the proposed metric's superiority.
2. The other main contribution of the paper is a model averaging procedure. However, the term "model averaging" is misleading, as the proposed procedure does not actually average model weights.
3. The experimental results in the paper are not convincing. It is not straightforward to conclude that "a larger value of S usually corresponds to a manifold learning outcome with a better visualization." From my perspective, tSNE is not worse than the "BEST METHOD" claimed in Figures 2, 3, and 4 of the paper.

Overall, the paper has some interesting ideas, but it needs further work to address these weaknesses.

---

> ### Author Response · Authors · 2023-05-04
>
> Thanks for your careful review. A section of Acknowledgement has been added in the end of the paper.  We would like to emphasize that our main contribution is to bring the model averaging idea to manifold learning. There are multiple existing manifold learning algorithms while there is no unanimous winner. To implement the model averaging which can enhance the robustness of manifold learning, we propose a new quality metric that is tuning-free and scale-invariant. Therefore, we have significantly revised the paper, which can be summarized as follows:
>
> 1.	We have reformulated the structure of the paper by first developing the model averaging procedure and then introducing the proposed metric that is particularly suitable for the model averaging. Therefore, a large amount of texts has been revised accordingly to restructure the paper.
> 2.	We have added Section 2, Related Work, to review briefly the model averaging and summarize the existing quality metrics for manifold learning.
> 3.	We have added two more quality metrics as comparison in Experiments to show superiority of our proposed quality metric. We have reported values of all quality metrics in all cases, as well as the optimal weights for computing MAML.
>
> Next, we provide the following detailed responses to your comments of Weaknesses and Requested Changes.
>
> Weaknesses:
>
> 1.	We have added two more tuning-free quality metrics for comparison: one is the proposed one with the Mahalanobis distance replaced by the Euclidean distance; the other is the average residual variance.
> For the synthetic data, the average residual variance often cannot reflect the visual quality, and the one using the Euclidean distance yields very different values for visually almost the same manifold learning outcomes. For the real data, the average residual variance is often not positively correlated with the classification accuracy. These observations are in contrast to those of the proposed metric, and thus indicate its superiority; please see the revised Experiment section for more details.
> 2.	We do not quite understand that “the proposed procedure does not actually average model weights”. The proposed method is to average multiple manifold learning outcomes from different algorithms, and the term “model averaging” comes from the literature of regression analysis as an alternative to model selection. Model averaging has been studied extensively in the statistical modeling literature. The method does not average model weights, while the weights are selected optimally to reflect the model fit. Good models receive higher weights while poor models receive lower weights. This can incorporate model uncertainties because no model is the true model in real applications and thus such an ensemble approach can enhance robustness of the results.
> 3.	Yes, we admit that visualization cannot be quantified numerically and different readers may possess different opinions. Therefore, we have replaced the conclusion
> "a larger value of S usually corresponds to a manifold learning outcome with a better visualization"
> by
> “A larger value of $\mathcal{S}$ usually corresponds to a manifold learning outcome that unwraps the surface with less distortion”.
>
> Requested Changes:
>
> 1.	As we mentioned earlier, we have added two more metrics for comparison. Our experiments are sufficient to demonstrate the proposed metric’s superiority over existing ones.
> 2.	As we mentioned earlier, we have changed the text “better visualization” by “… that unwarps the surface with less distortion”.

---

### Review · Reviewer_BNDR · 2023-04-09

**Summary Of Contributions:**

The paper proposes a quality metric to evaluate the learning results of manifold learning methods. The new metric is tuning-free and scale-invariant. The former is achieved by AUC of a neighbourhood agreement score, averaging over the neighbourhood size. The latter is achieved by replacing the Euclidean distance in the metric with the Mahalanobis metric. Based on the metric, the paper also develops an averaging scheme to linearly combine the (normalized) learned representations of multiple manifold learning methods. The averaged representation aims to maximize the proposed metric. Experiments on synthetic and real datasets demonstrate the utility of the metric and the averaging strategy.

**Audience:**

Yes

**Claims And Evidence:**

No

**Requested Changes:**

1. Add sufficient and detailed literature review of existing quality metrics for manifold learning.
2. Clarify the difference of the proposed metric with the exisiting ones.
3. Add experiments on showing why scale-invariance of the new metric is required. For example. this can be done by synthetic experiments where affine transformation is applied on the learned representations and metrics with and without using Mahalanobis distance are compared.
4. Compare the classification results obtained of MAML with the best manifold learning algorithm that produces the highest score.
5. Also it would be interesting to see whether the higher metric leads to higher classification results. It is desired to show the relation between the classification accuracy and the proposed quality metric.

**Strengths And Weaknesses:**

**Strengths**:
1. The use of Mahalanobis metric to ensure scale-invariance of the metric is interesting.
2. Experiments show promising results.

**Weaknesses**:
To the best of my understanding, the major claimed contributions are three-folds: (1) the proposed metric is tuning-free. (2) the proposed metric is scale-invariant. (3) a new model averaging strategy. I would like to list my concerns over each of the three points.
1. For the tuning-free aspect of the proposed metric, the use of AUC of the neighbourhood agreement score has already been studied, such as in [1,2]. It is unclear how the metric in this paper differs from the ones in [1,2].
2. It is unclear from the experiments how the scale-invariance of the new metric is better without comparing to the non-scale-invariant metrics (e.g. the ones using Euclidean distance for computing the agreement score).
3. It appears from the experiments that the model averaging approximates the best manifold learning method. In this case, why not just select the one that gives the highest score, instead of using an average. This is because sometimes averaging from various poor results may distort the learned shapes. For example, for the classification results, it may be better to just select the one which gives the highest score.


References:

[1] Lee, J. A., Renard, E., Bernard, G., Dupont, P., & Verleysen, M. (2013). Type 1 and 2 mixtures of Kullback–Leibler divergences as cost functions in dimensionality reduction based on similarity preservation. Neurocomputing, 112, 92-108.

[2] Lee, J. A., Peluffo-Ordóñez, D. H., & Verleysen, M. (2015). Multi-scale similarities in stochastic neighbour embedding: Reducing dimensionality while preserving both local and global structure. Neurocomputing, 169, 246-261.

---

> ### Author Response · Authors · 2023-05-04
>
> Thanks for your constructive review. A section of Acknowledgement has been added in the end of the paper. We would like to emphasize that our main contribution is to bring the model averaging idea to manifold learning. There are multiple existing manifold learning algorithms while there is no unanimous winner. To implement the model averaging which can enhance the robustness of manifold learning, we propose a new quality metric that is tuning-free and scale-invariant. Therefore, we have significantly revised the paper, which can be summarized as follows:
>
> 1.	We have reformulated the structure of the paper by first developing the model averaging procedure and then introducing the proposed metric that is particularly suitable for the model averaging. Therefore, a large amount of texts has been revised accordingly to restructure the paper.
> 2.	We have added Section 2, Related Work, to review briefly the model averaging and summarize the existing quality metrics for manifold learning.
> 3.	We have added two more quality metrics as comparison in Experiments to show superiority of our proposed quality metric. We have reported values of all quality metrics in all cases, as well as the optimal weights for computing MAML.
>
> Next, we provide the following detailed responses to your comments of Weaknesses and Requested Changes.
>
> Weaknesses:
>
> 1.	Indeed, the AUC has been used in Lee et al. (2015). The main difference between our metric and theirs is that we use the Mahalanobis distance while they use the Euclidean distance. However, the resulting evaluation outcome is improved dramatically; please see the revised section Experiments for more details.
> Moreover, one of our major contributions beyond Lee’s work is that we develop a model averaging procedure for manifold learning outcomes, which is new to the literature of manifold learning.
> 2.	Thanks for this constructive comment. We have added the quality metric using the Euclidean distance which corresponds to the one proposed by Lee et al. (2015). As shown in the new Table 1, the metric using the Euclidean distance yields very different values for visually almost the same manifold learning outcomes, while our proposed metric correctly matches visual results. This shows the importance of scale-invariance in assessing manifold learning outcomes.
> 3.	In statistical modelling, there are basically two choices: Selecting the best model using “model selection” or averaging results from all models based on “model averaging”. We cannot simply determine which one is better, because this is highly situation-dependent. Practitioners may prefer not to select a particular one but rather taking all algorithms into account because it is often unclear which one is the best (e.g., unsupervised learning). As to the classification experiments, they are intended to show the proposed metric is highly positively correlated with the classification results.
>
> Requested Changes:
>
> 1.	We have added Section 2, Related Work, to summarize the existing quality metrics for manifold learning.
> 2.	The main features of the proposed metric distinctive to the existing ones is that it is tuning-free and scale-invariant. The metric most similar to ours is the one in Lee et al. (2015), which has been added as comparison.
> 3.	We have added two more tuning-free quality metrics for comparison: one is the proposed metric with the Mahalanobis distance replaced by the Euclidean distance; the other is the average residual variance. It turns out that the current synthetic examples are sufficient to show the scale-invariance property is very important. For example, for the Swiss roll and the S shape datasets, the outcome produced by MAML is basically an affine transformed version of that produced by Isomap. Yet, the metric using the Euclidean distance yields distinct values for them. It demonstrates that the importance of scale invariance of a quality metric.
> 4.	We have added Table 5 to show the values of all quality metrics for the real datasets. It turns out that classification performances of MAML and the candidate algorithm that produces the highest $\mathcal{S}$ are indeed very similar.
> 5.	Comparing the current Table 3 and Table 5, we conclude that the value of the proposed metric is generally positively correlated with the classification accuracy. Specifically, for a given dataset with d = 2, we compute the correlation between the quality metric values and the classification accuracy averaged over k. The correlation of our proposed metric for the datasets COIL-20, MINST, and Chest X-ray are 0.94, 0.89, and 0.79, respectively. However, a higher value of the proposed metric does not ALWAYS lead to a better classification result. This is reasonable: Since a quality metric only assesses manifold learning outcomes and does not use any label information, we should not expect that a higher metric ALWAYS leads to a better classification result. We have summarized the discussion in the last paragraph of Section 5.2.

---

### Review · Reviewer_13H6 · 2023-04-20

**Summary Of Contributions:**

This paper proposes a quality metric based on the scale-invariant Mahalanobis distance. As applications of the proposed metric, the authors explore
1. Tuning parameter selection of manifold learning methods (such as the number of nearest neighbors in LLE)
2. Manifold learning model averaging
Experiments are conducted on both synthetic and real data.

**Audience:**

Yes

**Claims And Evidence:**

No

**Requested Changes:**

Please see the above section.

**Strengths And Weaknesses:**

** Strength **
1. The paper is easy to read. The fact that the proposed metric is tuning-parameter-free is interesting.
2. The experiment shown in Figure 1 suggests that the metric is useful (at least for this synthetic dataset) in choosing a good hyperparameter (number of nearest neighbors) of a given manifold learning algorithm.

** Weakness**
1. Without a detailed introduction and comparison to related works, it is not clear what is the contribution of the proposed paper in terms of the proposed quality metric. In section two, is the contribution (say, compared to Lee & Verleysen (2009)) only from changing the Euclidean distance to the Mahalanobis distance?
2. If it is only the distance change mentioned above, I would say the contribution of the paper is limited. The Mahalanobis distance is essentially the Euclidean distance after standardizing the representation, which of course becomes scale-invariant.
3. No computational issues have been addressed. For example, equation (2) requires the computation of pair-wise distance among all data points (for large K). This can be problematic for a large data set.
4. It is not clear what is rationale of "averaging a library of manifold learning models". Different models might produce drastically different output embedding, and taking a convex combination of the outputs as an averaged model seems weird. This is also shown in the result: the authors claim that the proposed model averaging algorithm (MAML) is able to produce an output embedding that is "visually close" to the best manifold learning algorithms from the library.  If this is the case, why not just choose the "best" manifold learning algorithm instead of taking the avarage?
5. It is not clear how one can solve the optimization problem (the unnumbered equation after 4) for model averaging. The problem is non-convex, non-smooth, and the evaluation of the objective function for each $w$ is already computationally expensive (see point 4).

---

> ### Author Response · Authors · 2023-05-04
>
> Thanks for your helpful comments. A section of Acknowledgement has been added in the end of the paper. We would like to emphasize that our main contribution is to bring the model averaging idea to manifold learning. There are multiple existing manifold learning algorithms while there is no unanimous winner. To implement the model averaging which can enhance the robustness of manifold learning, we propose a new quality metric that is tuning-free and scale-invariant. Therefore, we have significantly revised the paper, which can be summarized as follows:
>
> 1.	We have reformulated the structure of the paper by first developing the model averaging procedure and then introducing a new metric that is particularly suitable for the model averaging purpose. Therefore, a large amount of text has been revised accordingly to restructure the paper.
> 2.	We have added Section 2, Related Work, to review briefly the model averaging and summarize the existing quality metrics for manifold learning.
> 3.	We have added two more quality metrics for comparison in Experiments to show superiority of our proposed quality metric. We have reported values of all quality metrics in all cases, as well as the optimal weights for computing MAML.
>
> Next, we provide detailed point-by-point responses to the weaknesses.
>
> 1.	We have added Section 2, Related Work, to summarize the existing quality metrics for manifold learning. The metric most similar to ours is the one proposed in Lee et al. (2015), which has been added as comparison.
>
> 2.	Although it seems that the only difference is use of the Mahalanobis distance, the resulting evaluation outcome is improved dramatically. Specifically, we have added two more quality metrics for comparison: One is the proposed metric with the Mahalanobis distance replaced by the Euclidean distance; the other is the average residual variance. It turns out that scale-invariance is very important for assessing the manifold learning outcome. For example, for the Swiss roll and the S shape, the outcome produced by MAML is basically a rescaled version of that produced by Isomap. Yet, the metric using the Euclidean distance yields distinct values for them. It shows that using the Euclidean distance is not suitable for assessing manifold learning outcomes. Please refer to the revised section Experiments for more details.
> Moreover, one of our main contributions is that we develop a model averaging procedure for manifold learning outcomes, which is new to the literature of manifold learning.
>
> 3.	We agree that the computation issue may arise for comparing all pairwise distances which however is faced for almost all rank-based quality metrics. Our numerical studies did not reveal any practical difficulties of such computation given the high computing power nowadays.
>
> 4.	The rationale of averaging different manifold learning outcomes is similar to the rationale of model averaging used elsewhere, e.g., high-dimensional regression analysis: As an alternative to model selection, we may prefer to averaging over all possible candidate models instead of choosing a particular one, especially when we do NOT know which one is the best, e.g., unsupervised learning. It is then useful to combine different outcomes in a reasonable way which can enhance robustness of the method. Indeed, different models may produce drastically different output embeddings, and this is exactly the challenge that we manage to overcome. It is only reasonable to take a convex combination of different outcomes on the same scale, so it is necessary to rescale the different outcomes before averaging over them. To do this, we must develop a reasonable quality metric that is scale-invariant. Classification accuracy is only applicable in supervised learning, and the classification part in the paper is intended to show the proposed metric is highly positively correlated with the classification accuracy, indicating the validity of the proposed metric.
> Indeed, we may select the “best” outcome in some situations, while the model averaging procedure provides an alternative when it is not clear which one is the best. The MAML performs similarly to the candidate algorithm that is visually “the best” or leads to the most accurate classification, which exactly validates our proposed method. Please see the revised Introduction and Experiments for more details.
>
>
> 5.	We have added in the last paragraph of Section 3 that:
> “In the experiments, we use the built-in function fmincon with the SQP (sequential quadratic programming) algorithm in Matlab to implement the optimization procedure. Furthermore, we use multiple starting points to stabilize the result.”
> The code for reproducing the experimental results can be made publically accessible upon the acceptance of the paper.

---

> > ### Comment · Reviewer_13H6 · 2023-05-22
> > **Thank you for the detailed response**
> >
> > I thank the authors for their comprehensive response to my inquiries. The revised paper has undoubtedly enhanced my comprehension of this work's contributions.
> >
> > However, I still harbor some reservations regarding model averaging in the context of manifold learning. In regression, averaging can be perceived as a voting mechanism concerning a ground-truth target "value." Yet, the notion of taking a convex combination of multiple manifolds, even after normalization—many of which could potentially be mere rotated replicas of each other—seems less intuitive to me. Furthermore, I remain unconvinced by the authors' assertion that computational cost is a non-issue "given the high computing power nowadays." Since the provided datasets are all relatively small, it is hard to assess whether the proposed method, which involves optimizing over a non-trivial function (the proposed evaluation metric) whose function evaluation could already be computationally expensive.

---

> > > ### Author Response · Authors · 2023-06-06
> > > **Response to further comments**
> > >
> > > Thanks for reviewing our response and revised manuscript.
> > >
> > > Indeed, unlike supervised learning such as regression where there is ground truth, model averaging for manifold learning lacks a canonical metric to be optimized. This is well acknowledged in the paper. However, a convex combination of normalized manifold learning outcomes often yields competitive results in comparison to each individual outcome as shown in the paper. We believe such a new idea is potentially useful to make the manifold learning more robust as well as unifying different approaches.
> > >
> > > As to the computational issue for large datasets, we admit that it would be problematic when the sample size is extremely large. Comparing pairwise distances is indeed time costly but it is ubiquitous in constructing statistics or learning procedures. Such pairwise comparisons exist in all rank-based quality metrics and K-NN based manifold learning algorithms. For a concrete example, the Isomap cannot yield any result (the Isomap breaks down) in a reasonable time when the sample size reaches 58 thousand (McInnes et al. 2018). That is, a candidate algorithm may already fail to compute for a large dataset, so the model averaging procedure including that candidate algorithm becomes infeasible.

---

### Review · Reviewer_o362 · 2023-05-04

**Summary Of Contributions:**

The authors consider the problem of developing an evaluation measure for manifold learning methods. They proposed to base such a measure on ranks: first, they calculate ranks of neighboring points w.r.t. to considered points both for the original dataset and for the compressed dataset; second, they compare the sets of indices for each observation, obtained for the original and for the compressed representations. Then, they calculate an area under a curve, describing the dependence of the introduced normalized measure of similarity between the sets of indices on the number of the considered nearest neighbors.

The authors also proposed to consider an ensemble approach to combine different compressed representations from different manifold learning methods. First, they normalize the representations from the considered methods and then they weigh them to obtain an average representation by optimizing the proposed quality measure w.r.t. to the weights.

The authors demonstrated how the method works on several toy and real-world datasets. To estimate the quality of manifold learning (ML) they used the same quality measure which we optimize when applying MAML. Also, they provided results of classification based on compressed data representations.

**Audience:**

Yes

**Broader Impact Concerns:**

I do not think that there are any ethical concerns.


**Claims And Evidence:**

No

**Requested Changes:**

There are comments in the weaknesses subsection above. First of all, the authors should take this into account in their work.


**Strengths And Weaknesses:**

Strengths:
1) the topic of the paper is important
2) the authors considered some interesting ideas that can be useful for applications.
In particular, the proposed idea to use ranks to compare data structures in the original and compressed spaces sounds interesting.

Weaknesses:
1) the main weakness of the paper is the considered experimental protocol:

a) the authors compared different manifold learning methods with the MAML procedure using the same quality measure that they optimized in the MAML procedure.
What is about the out-of-sample extension/generalization ability of the considered manifold learning methods?
Also, it looks like there is a bias when comparing manifold methods with MAML based on a quality measure, which is directly optimized in the case of the MAML procedure.

b) the proposed quality measure is a contribution, so it should be compared with other standard approaches to estimate the quality of manifold learning methods. E.g.
- In https://proceedings.mlr.press/v162/barannikov22a.html the authors consider approaches to compare neural network representations such as CKA, CCA-like measures, etc. Also, they proposed a topology-based approach (RTD). Such approaches are also rather standard when estimating the quality of ML methods.
- In https://openreview.net/forum?id=lIu-ixf-Tzf the authors compare original data with latent representations by (1) linear correlation of pairwise distances, (2) Wasserstein distance (W.D.) between H0 persistence barcodes (Chazal & Michel, 2017), (3) triplet distance ranking accuracy (Wang et al., 2021) (4) RTD. These are other examples of quality measures.
- In https://lvdmaaten.github.io/publications/papers/TR_Dimensionality_Reduction_Review_2009.pdf the authors made a comparative review of manifold learning methods and used the ‘trustworthiness’ and the ‘continuity’ of the low-dimensional embeddings.

Does the quality measure proposed in the paper allow for properly ranked results of different manifold learning methods compared to other quality measures? At least using artificial data examples it is possible to get some ground truth ranking of compared manifold learning methods and then compare this ranking with ranking based on a considered quality measure.

Also, in the papers, mentioned above, the authors consider more diverse artificial and real datasets for testing their methods. I would propose to use these additional datasets.

c) In MAML the authors optimize the quality measure w.r.t. weights. I propose to provide some visualisation of the resulting optimal weights. In some examples, the value of the optimized quality measure for MAML is almost the same as the quality measure value for some other manifold learning methods. This looks like in MAML this particular manifold method dominates other manifold methods in the ensemble. Can we consider MAML as a model selection tool?
Also, is there any qualitative explanation of the situation when weights for different manifold methods are comparable and so as a result we get a truly averaged latent representation? Why is linear averaging of different latent representations mathematically correct? It seems that different ML methods have different latent coordinate systems which are not comparable. Could the authors justify somehow that it is really enough to use a standard linear transformation (3) to compare and combine the outcomes obtained from distinct ML methods?

2) The author mentioned that “Our proposed model averaging procedure is thus the first attempt to obtain a unified outcome from various manifold learning algorithms.” Then what do the authors think about the papers:
- https://link.springer.com/chapter/10.1007/978-3-642-39678-6_16
- https://link.springer.com/chapter/10.1007/11881070_5

3) When ML methods are used as feature extractors for classification, it is important that these ML methods have an out-of-sample extension. At the same time, it is not clear whether MAML has this capability.

4) The authors did not specify how they performed testing of the classifiers. Did they compress simultaneously train and test datasets? Or for test datasets, they somehow calculated out-of-sample extensions?

---

> ### Author Response · Authors · 2023-05-09
>
> Thanks very much for your detailed comments. We didn’t expect that there was one more review coming after a few hours when we submitted our revision. Let us first stress the main changes for the last revision:
>
> 1.	We would like to emphasize that our main contribution is to bring the model averaging idea to manifold learning, so we have reformulated the structure of the paper by first developing the model averaging procedure and then introducing a new metric that is particularly suitable for the model averaging purpose. Therefore, a large amount of text has been revised accordingly to restructure the paper.
> 2.	We have added Section 2, Related Work, to review briefly the model averaging and summarize the existing quality metrics for manifold learning.
> 3.	We have added two more quality metrics for comparison in Experiments to show superiority of our proposed quality metric. We have reported values of all quality metrics in all cases, as well as the optimal weights for computing MAML.
>
> Next, we response to your comments of Weaknesses:
>
> 1.
>
> a)	Since the proposed MAML is a weighted sum of candidate manifold learning outcomes, the out-of-sample extension/generalization ability of MAML mainly depends on that of the candidate manifold learning algorithms, especially the one that receives the largest weight. You are totally correct that it is unfair to compare MAML with other manifold learning algorithms using the proposed quality metric, because MAML is computed by optimizing the proposed metric. This is not what we intend to show. In the old table 1, we only mean to show that higher values of our proposed metric usually correspond to a manifold learning outcome that unwraps the surface with less distortion. Now the new table 1 also include other two quality metrics.
>
> b)	Yes, we have added two more quality metrics for comparison now. To show the importance of scale invariance, we have considered the quality metric with the Mahalanobis distance replaced by the Euclidean distance. We agree that there are more quality metrics that are reasonable in other contexts, and we do not intend to beat them in all contexts. The main purpose of the proposed quality metric is to develop the model averaging, and thus we believe the current experimental results are sufficient.
>
> c)	Yes, we have already added Tables 2 and 4 to show the resulting optimal weights. Indeed, most of the weights are quite close to one, indicating that MAML particularly favors that candidate algorithm. If the weight assigned to a candidate algorithm is exactly one, then we can regard MAML as a model selection tool. However, ''model selection'' and ''model averaging'' are exactly two complementary tools developed in the literature. That is, these two terminologies are somehow antonymous. We have added more discussion on this in Sections 1 and 2. As to theoretical justification of MAML, it warrants future research.
>
> 2.	Thanks for providing these two papers. The first one Zhang et al. (2013) is indeed interesting. The authors proposed to rotate and rescale a standardized manifold learning outcome such that it is close to another manifold learning outcome. We have now cited it and revised the claim in Introduction. The second one Zhang et al. (2006) is less relevant to our paper. They focus on combine different classifiers induced by local linear embedding and linear discriminant analysis instead of combing different manifold learning outcomes.
>
> 3.	As stated earlier, the out-of-sample extension/generalization ability of MAML mainly depends on that of the candidate manifold learning algorithm that receives the largest weight.
>
> 4.	Yes, we applied the manifold learning algorithms for all the data first and then divided them into training and testing datasets. This has been now clearly specified in Section 5.2.

---

### Decision · Action_Editors · 2023-06-08

**Recommendation:** Reject

**Comment:**

Because of the above points (in claims and evidences), all the reviewers as well as this AE think that the paper in its current form cannot be accepted to TMLR. It does not meet the criteria of the paper should have clear evidence to support its two claims.



**Audience:**

Yes, the idea is interesting to some of TMLR's audience.

**Claims And Evidence:**

The paper has two main contributions: 1) averaging of the scaled coordinates (where scaling is by the inverse of the covariance matrix) and 2) quality metric definition S(X, Z) that is scale-invariant and that measures how good are the learned low-dimensional embeddings Z are with respect to the original embeddings X.
All the reviewers have appreciated the efforts, but have also highlighted the key issues with these contributions. Below are the main points of contention.

1. When generating the scaled coordinates (Section 3) z from y, we use the scaling z = V^{-1/2} y. I don’t have an issue with the choice V, but why is that a good choice?

2. Now the new coordinates “z” from different algorithms may lie in different d-dimensional spaces (what is the guarantee that they are in the “same” space, e.g., they may be rotated?). So why is plain averaging (equation 2) a good idea?

3. Reviewer 0362 is absolutely spot on in highlighting the above issues. Ideally, we should ensure that all the new embeddings z from different algorithms should lie in the same space, e.g., there should be Q_k such that z_k = Q_k V_k^{-1/2} y_k, where Q_k are such that the {z_k} are all rotated properly to be in the same space. (

4. A glaring issue: Equation 1 is not rotationally invariant, i.e., if y gets rotated by Q then z also gets rotated by Q. Overall, y —> Qy would lead to z —-> Qz, please verify this! So, even if the averaging procedure of equation 2 works, I can make it super **bad** by arbitrarily rotating the embeddings of different models. So, equation 2 is not rotationally invariant! And therefore, vanilla averaging is incorrect.

5. Now coming to the second contribution which is on defining S, there is no proper theoretical justification for the same. I would have hoped that the paper did a good and rigorous justification for the same, theoretically and empirically.

6. The empirical results are not convincing either. What should be gathered from Tables 1 and 3? Does averaging work or not work?